# Using Sensors for Player Development: Assessing Biomechanical Factors Related to Pitch Command and Velocity

**DOI:** 10.3390/s22218488

**Published:** 2022-11-04

**Authors:** Cristine Agresta, Michael T. Freehill, Bryson Nakamura, Samuel Guadagnino, Stephen M. Cain

**Affiliations:** 1Department of Rehabilitation Medicine, University of Washington, Seattle, WA 98195, USA; 2Department of Orthopedic Surgery, Stanford University, Redwood City, CA 94063, USA; 3Stanford Baseball Science Core, Stanford University, Redwood City, CA 94063, USA; 4Department of Chemical and Biomedical Engineering, West Virginia University, Morgantown, WV 26506, USA

**Keywords:** throwing, wearables, pitching, kinematics, accelerometers, skill, velocity

## Abstract

Pitching biomechanical research is highly focused on injury prevention with little attention to how biomechanical data can facilitate skill development. The overall purpose of this study was to explore how sensor-derived segment kinematics and timing relate to command and ball velocity during baseball pitching. We used a cross-sectional design to analyze a series of pitches thrown from 10 collegiate baseball pitchers. We collected biomechanical data from six inertial sensors, subjective command from the pitchers, and ball velocity from a radar device. Stepwise regression analyses were used to explore biomechanical variables associated with command for all pitches and ball velocity for fastballs only. We found that only peak forearm linear acceleration was significantly associated with command, whereas several segment kinematic measures were significantly associated with ball velocity. Our results suggest that different biomechanical variables are linked to specific pithing skills. Our findings suggest that end-effector (forearm) movement is more important for pitch command, whereas proximal-to-distal (pelvis, trunk, upper arm, forearm) segmental movement is important for ball velocity.

## 1. Introduction

Baseball pitchers throw a repertoire of pitch types (e.g., fastball, curveball, slider) during games. Developing a new pitch may take years and the length of development differs across player and pitch type. During this developmental period, pitchers may be susceptible to undue injury or fatigue because of the repetitive throwing. Thus, finding a solution to help coaches improve the efficiency and effectiveness of pitch development is desirable.

Pitching biomechanics research is largely interested in associations or predictors of arm injury [1,2,3,4] with little emphasis on how biomechanical data can facilitate skill development. In studies that investigate skill, generation of ball velocity is the priority outcome [5,6,7,8,9]. However, the ability to throw the ball as it is intended to move (i.e., pitch command) is arguably just as important a skill to develop. Command involves a more nuanced level of control, as the pitcher is not just trying to throw strikes, but to execute a specific location of the ball in or out of the strike zone, with the goal of keeping each pitch out of the middle of the plate [10]. Identifying the primary biomechanical factors influencing command and ball velocity can facilitate skill development.

The method of data capture must be considered when assessing skilled movement. Data capture must be continuous and collect consecutive pitches. Additionally, capture should occur in real settings to assess how pitchers adapt movement patterns to training. The capture method must be portable, affordable, and easy-to-use by coaches and not significantly disrupt the flow of practice. Inertial sensors offer a solution to capture needs. Despite this, pitching biomechanics are most frequently captured in a lab with optical motion capture [11,12]. In this setting, only a few non-consecutive pitches are captured (e.g., 3 pitches thrown with highest velocity recorded as strikes) and analyzed [3,13,14] or used in machine learning models for prediction of ball velocity [15].

Because there is not a strong understanding of sensor-derived biomechanical factors related to pitching skill, the purpose of this pilot study was to explore how sensor-derived segment kinematics and timing relate to command and ball velocity during baseball pitching. Although the data is preliminary, we hypothesized that biomechanical variables associated with pitch command would differ from those associated with ball velocity.

## 2. Materials and Methods

This study was approved by the Institutional Board Review. All participants provided informed consent before data collection.

### 2.1. Participants

Data were collected from 10 healthy male collegiate pitchers (left-handed = 4, mean age = 19.2 ± 1.2 years, height = 186.4 ± 6.2 cm, mass = 86.5 ± 10.5 kg). All participants were cleared for practice and free of injury.

### 2.2. Data Collection and Analysis

Each pitcher wore 6 inertial sensors (Opal, APDM, Inc., Portland, OR, USA) during their regularly scheduled bullpen session (Figure 1). Pitchers threw from regulation distance on the same regulation mound in their team indoor cage. Pitchers threw a series of approximately 35 pitches to a live catcher in a pre-determined order set by the pitching coach. Each pitcher threw at least 18 fastballs, 7 change-ups, and 10 breaking balls (curveballs, sliders, or cut fastballs). Pitchers were instructed to throw at their full effort since data were collected in preseason (October) and pitchers had been throwing for several weeks. Five pitchers threw 1–4 additional pitches beyond the set order per self-request or on the coach’s recommendation. The pitchers had intended areas of the strike zone to locate the pitch as determined by the pitching coach. The locations included (1) box, (2) glove-side, and (3) arm-side. Box means the pitch should land in the middle of the strike zone. Glove-side means the pitch should land to the side corresponding to their glove hand. Arm-side means the pitch should land to the side corresponding to their throwing arm. Pitch type and location were verbalized by the pitching coach and manually checked against the predetermined order. All throws/pitches were recorded using the inertial sensors.

A 5-point Likert scale was used for subjective command. Pitchers were asked to verbally rate “How well did you execute the intended pitch?” immediately after each pitch in the bullpen session. A score of “1” indicated the lowest command level, and a score of “5” indicated the highest command level. We explained the scoring system to the pitchers before the start of the bullpen session and they were reminded how to score as needed. Due to the nature of bullpen training, we did not ask the pitcher to elaborate on what aspects of command they were assessing. Pitchers did not review pitch performance data, so scores were not influenced by objective data.

Ball velocity (miles per hour), horizontal and vertical break, and x-y strike index coordinates were the pitch performance variables captured using a consumer-available radar unit (Pitching 2.0, Rapsodo, Inc., Brentwood, MO, USA). The device was calibrated, per manufacturer specification, and placed approximately 15′6” from the front of home plate, facing the pitcher. Operational definitions are given in Table 1 and are taken directly from the manufacturer’s manual (Rapsodo, Inc. (2019). Pitching 2.0: User Manual. See Appendix A).

Sensors were placed on the dorsum of each foot, the throwing forearm, the throwing upper arm, the distal sternum, and the pelvis. Each sensor incorporated a tri-axial accelerometer (range: ±200 g) and a tri-axial gyroscope (range: ±2000 deg/s) and sampled at 512 Hz. We measured segmental angular velocity, linear acceleration, angular acceleration, or orientation of the pelvis, trunk, and throwing arm from the sensors. Sensors were affixed before the start of warm-up to allow for a seamless transition from warm-up exercises and throws into pitching. Sensor data collection was initiated immediately before the start of warm-up and was collected continuously until the end of the bullpen session. A time marker was added to the data stream during collection to denote the transition from warm-up to the start of the bullpen session. One pitch type per player was recorded via high-speed video. High-speed video (AOS Technologies, Plymouth, MI, USA) was utilized in postprocessing, if needed, to verify segment motion derived from sensor data.

We examined angular velocity and angular acceleration of the pelvis and trunk about the vertical axis, along with throwing upper and forearm linear acceleration, because ball velocity is generated through segmental power transfer [16,17] or because of the association with ball velocity [3]. Likewise, we examined the timing of segmental kinematics to provide a crude estimate of movement coordination, which has been shown to influence ball velocity [4,9]. We examined throwing arm orientation and foot orientation at foot strike, stride time, and peak rotational separation of the trunk to pelvis (trunk separation), as they are often considered important variables by coaches for pitch performance [11,13,18,19]. Calculated biomechanical variables are listed and defined in Table 1.

### 2.3. Statistical Analysis

Stepwise regression analyses were used to help identify a list of potential explanatory variables related to pitch command and ball velocity (fastballs only). Biomechanical variables included in the model are listed in Table 1. Alpha was set a priori at 0.01.

A Spearman’s rank correlation coefficient test was used to assess the relation between pitch command score and vertical break, horizontal break, and x-y strike index coordinates for each intended pitch location (box, glove-side, arm-side). Strong associations were 0.9 to 0.7, moderate associations were 0.4 to 0.6, and weak associations were <0.4 [20].

## 3. Results

Table 2 displays the biomechanical variables that remained for pitch command (all pitch types) and ball velocity (fastballs only) following backward elimination stepwise regression. Forearm peak resultant acceleration was associated with pitch command and 10 of the 15 variables measured were related to ball velocity.

Only vertical break was significantly associated with level of command for all three intended pitch locations—box (rho = 0.21, *p* = 0.008), glove-side (rho = 0.41, *p* = 0.001), and arm-side (rho = 0.32, *p* = 0.010) (Figure 2). Associations were weak for box and arm-side locations and moderate for the glove-side location.

## 4. Discussion

The purpose of this pilot study was to explore how sensor-derived segment kinematics and timing relate to command and ball velocity during baseball pitching. In support of our hypothesis, we found that biomechanical variables linked to pitch command differed from variables linked to ball velocity. Our findings suggest that having command and generating ball velocity are separate skills with their own biomechanical associations. Only peak forearm resultant linear acceleration explained differences in levels of pitch command, whereas 10 variables across multiple segments helped to explain differences in ball velocity.

To date, only one study has looked at a similar skill to command—pitch control. In this study, pitch location consistency in relation to biomechanical variability at the instant of foot contact and maximal shoulder external rotation was used to determine control [20]. We chose to use pitch command rather than using an objective measure of pitch location consistency because subjective command better represents the construct, which is a self-perceived feeling of movement of the ball through a desired trajectory. Command is more aligned with player development, as pitchers need to refine movement patterns to improve command. Because command and control are related and there is no existing biomechanics literature on command, we compared our results to Glanzer et al. [21]. Glanzer et al. [21] found less biomechanical variability to be associated with greater pitch location consistency. Differences between their findings and ours can be attributed to how the outcome metric was conceptualized and calculated, differences in pitch sample sizes, and selection of throwing biomechanics. First, our study used subjective command rather than an objective strike zone point. The pitchers in our study were given a broad location to aim for (box, glove-side, arm-side), which provided a regional target rather than specific (x-y) point. Second, Glanzer et al. [21] used only a small number of pitches (8 to 10 for each pitcher) for their analyses and only one pitch type (fastballs), whereas we used approximately 35 pitches and varying pitch types per player. It may be that their dataset was too small to demonstrate the native variability that exists in throwing biomechanics. Finally, although we did report on forearm and upper arm orientation at foot contact, most of our variables described timing or peak accelerations and velocities. We also used the full trajectory of segment kinematics in calculation, rather than being restricted to a specific event (e.g., foot contact, maximal shoulder external rotation). These differences in calculations may have contributed to differences in findings.

We performed ad hoc analyses to assess biomechanical variability and the correlation of command to strike zone coordinates. We found a large degree of variability in biomechanical variables assessed (Table 3), except for forearm and upper arm peak resultant linear acceleration and time of ball release relative to foot strike. This was expected, as multi-joint skilled movements typically display greater proximal segment motor flexibility (as seen through greater variability of motion) and less end-effector variability [22,23,24]. Throwing arm linear acceleration and timing of ball release may be significant features needed for locating the pitch or creating a specific ball trajectory. Pitch command was weakly-to-moderately associated with vertical break position but not x-y coordinates in the strike zone (Table 4).

Our findings for variables explaining ball velocity support the kinetic chain principle [25], where energy travels along rotating sequential body segments and transfers to the ball for linear velocity. Our findings also match previous work where a relationship exists between ball velocity and the power and timing of trunk rotation [16], peak absolute pelvis angular acceleration [8], velocity relative to foot contact [7], and the efficiency of the kinetic chain [8,9,16]. Five of the ten explanatory variables relate to the kinematics and timing of the trunk or pelvis. Previous work [17] found that the onset of peak trunk rotation occurred later in the pitch cycles in more skilled players. Fleisig et al. [13] found that more advanced pitchers displayed higher upper torso velocities than less skilled players and also produced a significantly higher ball velocity. We also found that segment kinematics and timing of the torso and pelvis were associated with ball velocity. 

This study has several limitations. First, we analyzed only a small set of pitchers with the same general skill level and age range. A larger cohort across various skill levels may yield different results. Second, we only collected cross-sectional data, so we cannot comment on how pitchers develop pitch command or ball velocity. Third, we selected biomechanical variables that were able to be captured accurately using inertial sensors and were robust to changes in sensor placement and algorithm choice to enable repeatable data collections in a real-world setting. It is quite possible that additional variables captured using different instrumentation may better explain command or ball velocity. Last, subjective command was verbalized aloud by the pitcher in the presence of coaching staff and teammates. It is possible that some pitchers did not report a true estimation of their perceived pitch command for some pitches. These exploratory findings should serve as a starting point to using sensor-derived biomechanics for skill development. Future research should increase and include biomechanical measures in relation to pitching skill, use sensors to monitor athletes over time as they develop in skill level, and analyze pitchers with varying skill levels.

## 5. Conclusions

Pitch command and generation of ball velocity represent two separate performance attributes with distinct biomechanical contributions. Our findings suggest that ball velocity is a product of whole-body segmental motion acting to transfer energy from proximal to distal segments and then to the ball. In contrast, subjective pitch command appears to be focused on end-effector motion. Coaches and players may benefit from using sensors to monitor distal segment kinematics when developing command and multiple segment kinematics when developing ball velocity.

## Figures and Tables

**Figure 1 sensors-22-08488-f001:**
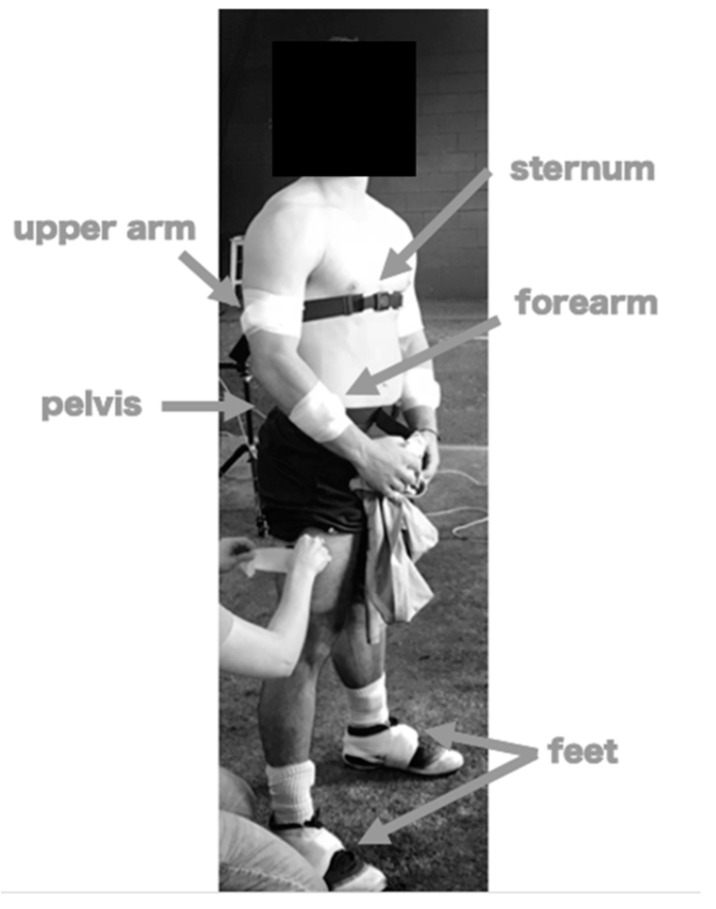
Sensor placement of six inertial sensors (Opal, APDM, Inc., Portland, OR, USA) affixed using straps or adhesive tape.

**Figure 2 sensors-22-08488-f002:**
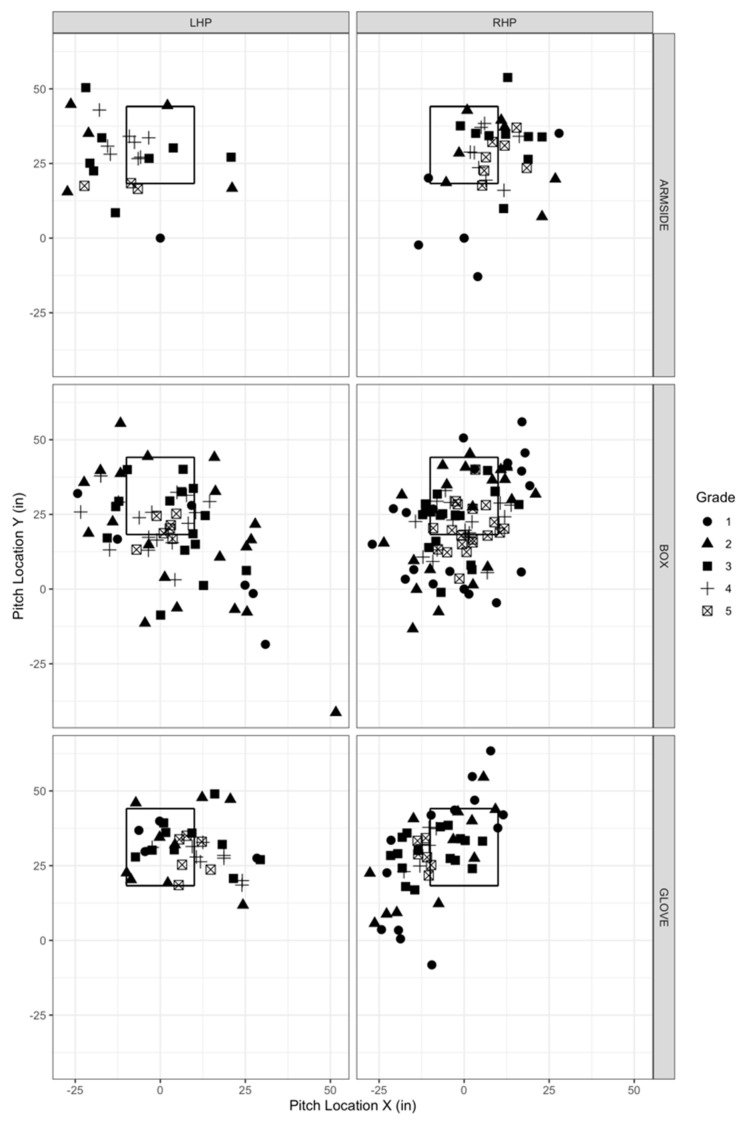
Pitch locations for left-hand pitchers (LHP) and right-hand pitchers (RHP). Pitch locations are shown from the pitcher’s perspective. Three intended locations (box, arm-side, and glove-side) are shown separately for LHPs and RHPs. The square represents the strike zone; coordinates to construct were taken from Pavlidis, Judge, & Long, 2017 [10]. Points (*n* = 1) where the strike Y was greater than 25 were removed for better visualization. This point remained in data analysis.

**Table 1 sensors-22-08488-t001:** Descriptions of biomechanical and pitching variables of interest. Biomechanical variables were calculated from inertial sensor data and are described below. Pitch variables were taken from the Rapsodo output and are described below using information provided in the Rapsodo user manual (Rapsodo, Inc. (2019). Pitching 2.0: User manual).

Biomechanical Variables	Description
Segmental Position	
Orientation of forearm relative to vertical at foot strike (°)	Forearm sensor orientation (approximately the angle of the long axis of the forearm) relative to vertical at the instant of foot contact
Orientation of upper arm relative to vertical at foot strike (°)	Upper arm sensor orientation (approximately the angle of the long axis of the upper arm) relative to horizontal at the instant of foot contact
Foot orientation at foot contact (°)	Pitch orientation of the foot at the instant of foot contact in degrees, where −90 is straight down, 0 is horizontal, and 90 is straight up
Peak rotational separation of the torso and pelvis (°)	Maximum angular difference between the rotation of the torso about vertical and the pelvis about vertical
Segmental Angular Velocity/Acceleration or Linear Acceleration	
Peak linear acceleration of the forearm (m/s^2^)	Peak forearm linear acceleration magnitude during the pitch cycle
Peak linear acceleration of the upper arm (m/s^2^)	Peak upper arm linear acceleration magnitude during the pitch cycle
Peak rotation rate of the torso about vertical (°/s)	Maximum rotation rate of the torso about a vertical axis during the pitch cycle
Peak rotation rate of the pelvis about vertical (°/s)	Maximum rotation rate of the pelvis about a vertical axis during the pitch cycle
Peak pelvis angular acceleration about vertical (°/s^2^)	Maximum angular acceleration of the pelvis about a vertical axis during the pitch
Peak torso angular acceleration about vertical (°/s^2^)	Maximum angular acceleration of the torso about a vertical axis during the pitch
Segmental Timing	
Time of ball release relative to foot strike (s)	Estimated time of ball release relative to the instant of lead foot contact
Stride time (s)	Time from when the lead foot first begins moving forward during the striding phase of the pitch cycle to instant of foot contact
Time of peak upper arm acceleration relative to foot strike (s)	Time from the instant of foot contact to peak upper arm acceleration
Time of peak rotational separation of the torso and pelvis (s)	Time of the peak rotational separation of the torso and pelvis relative to foot strike
Time of peak pelvis rotational acceleration relative to foot strike (s)	Time of the peak angular acceleration of the pelvis about a vertical axis relative to the instant of foot contact
Time of peak torso rotational acceleration relative to foot strike (s)	Time of the peak angular acceleration of the torso about a vertical axis relative to the instant of foot contact
Time of peak pelvis rotation rate relative to foot strike (s)	Time of the peak rotation rate of the pelvis about a vertical axis relative to the instant of foot contact
Time of peak torso rotation rate relative to foot strike (s)	Time of the peak rotation rate of the torso about a vertical axis relative to the instant of foot contact
Pitch Performance Variables	
Vertical break	How much the ball is moved vertically when it crosses the strike zone compared to what its position would have been without spin
Horizontal break	How much the ball is moved horizontally when it crosses the strike zone compared to what its position would have been without spin
Strike index (*x*-axis)	The horizontal position of the ball in the strike zone
Strike index (*y*-axis)	The vertical position of the ball in the strike zone
Velocity (mph)	How fast a pitch is traveling during flight
Spin rate (rpm)	The rate at which the ball spins during flight

Note: The pitch cycle is defined to begin with the first movement (lifting of the lead foot) and end with the trail foot touching the ground after ball release. Negative times indicate that an event occurred before foot contact, whereas positive times indicate events that occurred after foot contact. mph = miles per hour; rpm = revolutions per minute.

**Table 2 sensors-22-08488-t002:** Biomechanical variables remaining from stepwise regression model using backward elimination. Variables are separated by pitch performance concept: command for all pitch types and ball speed for fastballs only.

Biomechanical Variable	Standardized Coefficient (*β*)	Standard Error	*p*-Value (CI)
*Command*			
Forearm peak resultant acceleration (m/s^2^)	0.008	0.003	0.010 (0.002–0.001)
*Ball velocity* (mph)			
Time of peak rotational separation of the torso and pelvis (s)	3.82	1.56	0.016 (0.73–6.91)
Peak acceleration of the forearm (m/s^2^)	0.01	0.00	<0.001 (0.01–0018)
Peak rotation rate of the torso about vertical (°/s)	0.03	0.00	<0.001 (0.03–0.04)
Peak rotational separation of the torso and pelvis (°)	0.29	0.03	<0.001 (0.24–0.34)
Orientation of upper arm relative to vertical at foot strike (°)	0.16	0.02	<0.001 (0.11–0.20)
Peak torso angular acceleration about vertical (°/s^2^)	0.00	0.00	<0.001 (0.001–0.002)
Peak rotation rate of the pelvis about vertical (°/s)	0.01	0.00	0.002 (0.01–0.02)
Time of peak pelvis rotation rate relative to foot strike (s)	−62.59	21.84	0.005(−105.71–−19.47)
Orientation of forearm relative to vertical at foot strike (°)	−0.06	0.01	<0.001 (−0.09–−0.03)
Time of ball release relative to foot strike (s)	−66.15	27.36	0.017 (−12,017–−12.13)

mph = miles per hour; CI = confidence interval.

**Table 3 sensors-22-08488-t003:** Pitch performance and biomechanical variables (Mean ± SD) across self-reported levels of command (1–5). A score of 1 indicates the lowest self-reported command and a score of 5 indicates highest self-reported command. See Table 1 for descriptions of biomechanical variables.

	1	2	3	4	5
Orientation of forearm relative to vertical at foot strike (°)	28.1 ± 21.4	26.7 ± 28.4	28.5 ± 26.7	29.3 ± 27.1	31.4 ± 26.1
Orientation of upper arm relative to vertical at foot strike (°)	6.6 ±10.6	3.9 ± 15.8	4.4 ± 16.6	−0.1 ± 17.8	5.04 ± 16.1
Foot orientation at foot strike (°)	6.9 ± 6.8	5.9 ± 7.46	6.92 ± 7.78	6.03 ± 6.8	7.1 ± 6.2
Peak rotational separation of the torso and pelvis (°)	39.7 ± 23.6	38.7 ± 25.2	35.7 ± 20.9	37.8 ± 22.7	34.7 ± 22.8
Peak acceleration of the forearm (m/s^2^)	1237.9 ± 172.5	1319.8 ± 214.0	1314.5 ± 249.0	1349.5 ± 240.0	1345.7 ± 179.9
Peak acceleration of the upper arm (m/s^2^)	1027.4 ± 175.6	1025.0 ± 189.0	1021.7 ± 172.9	1021.7 ± 172.9	1083.0 ± 222.7
Peak rotation rate of the torso about vertical (°/s)	891.9 ± 300.9	877.3 ± 305.1	920.9 ± 259.8	897.4 ± 299.6	898.6 ± 262.8
Peak rotation rate of the pelvis about vertical (°/s)	736.7 ± 72.7	740.0 ± 121.4	747.4 ± 117.6	759.1 ± 138.4	735.1 ± 93.7
Peak pelvis angular acceleration about vertical (°/s^2^)	6624.8 ± 1295.9	6503.0 ± 1641.0	6905.8 ± 1642.2	6825.3 ± 751.5	6477.2 ± 1267.1
Peak torso angular acceleration about vertical (°/s^2^)	8534.1 ± 3105.2	8803.8 ± 3672.2	9223.0 ± 3333.6	9219.8 ± 4062.5	8832.1 ± 3135.0
Time of ball release relative to foot strike (s)	0.2 ± 0.1	0.16 ± 0.01	0.2 ± 0.01	0.2 ±0.01	0.2 ± 0.01
Stride time (s)	0.5 ± 0.2	0.5 ± 0.1	0.5 ± 0.2	0.5 ± 0.1	0.4 ± 0.4
Time of peak upper arm acceleration relative to foot strike (s)	0.2 ± 0.02	0.2 ± 0.02	0.2 ± 0.02	0.2 ± 0.02	0.2 ± 0.02
Time of peak rotational separation of the torso and pelvis (s)	0.002 ± 0.05	−0.01± 0.1	−0.01 ± 0.1	−0.02 ±0.2	−0.05 ± 0.2
Time of peak pelvis rotational acceleration relative to foot strike (s)	−0.03 ± 0.04	−0.04 ± 0.03	−0.03 ± 0.03	−0.04 ± 0.04	−0.04 ± 0.03
Time of peak torso rotational acceleration relative to foot strike (s)	−0.07 ± 0.2	−0.06 ± 0.2	−0.04 ± 0.2	−0.06 ± 0.2	−0.05 ± 0.2
Time of peak pelvis rotation rate relative to foot strike (s)	0.05 ± 0.01	0.04 ± 0.02	0.04 ± 0.02	0.04 ± 0.02	0.04 ± 0.02
Time of peak torso rotation rate relative to foot strike (s)	0.02 ± 0.1	−0.002 ± 0.2	0.02 ± 0.2	−0.01 ± 0.2	0.02 ± 0.1

**Table 4 sensors-22-08488-t004:** Description (mean ± SD) of pitch performance measures of pitch movement (vertical and hori-zontal break) and location (x-y strike index coordinates) across levels of pitch command.

	Pitch Command Level				
	1	2	3	4	5
Vertical break	2.6 ± 8.1	4.3 ± 9.3	5.8 ± 8.8	7.6 ± 8.6	7.3 ± 11.0
Horizontal break	−1.2 ± 7.3	−0.5 ± 8.9	0.7 ± 9.2	−0.1 ± 10.9	−0.9 ± 10.5
Strike index (*x*-axis)	−0.5 ± 15.5	0.7 ± 16.1	−0.3 ± 13.1	−0.9 ± 10.8	0.8 ± 8.8
Strike index (*y*-axis)	20.3 ± 20.9	24.5 ± 18.6	23.9 ± 13.3	24.0 ± 9.5	21.7 ± 8.2

## Data Availability

The data that support the findings of this study are available from the corresponding author upon reasonable request.

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
