# Peer review of "Using Sensors for Player Development: Assessing Biomechanical Factors Related to Pitch Command and Velocity"

_sensors, 2022, doi:10.3390/s22218488_

Round 1

Reviewer 1 Report

Dear authors

The corrections comments are based on the manuscript.

Abstract

Concern # 1: the abstract Reorganize the abstract to conclude:

(a) The overall purpose of the paper and the research problems you investigated should be short.

(b) The basic design of the study.

(c) Major findings or trends found as a result of the study.

(d) A brief summary of your interpretations and conclusions.

Concern # 2: If the term is mentioned once in the introduction, there is no need to use abbreviations. In addition, when using the abbreviation, it must be written completely the first time.

Keywords

Concern # 2: Try to include additional keywords

Introduction

Concern # 3: The paper needs to clarify the motivation, challenges, contribution, objective, significance, and others. All the information should be presented in sequence idea. There is a need to answer the following questions:

Q1: How can you evaluate the presented results according to other studies?’ Prove that the literature review lacks such a study by more modern references.

Q2: ‘What is the importance of the presented paper?’

Q3: What is the main challenge and issues in this study?

Q4: What is the criticism and gap analysis for academic literature that attempts to provide a solution?’

Q5: What are the recommended solutions for such challenges and their issues?’

Q6: What are the present study's implications, contributions, and novelty?’

Concern # 3: Explain the importance of Player Assessing by adopting it in many fields and applications. I suggested citing several previous works that used AI to solve different problems Player Assessing as a literature review.

Methodology

Concern # 4: the proposed method is so confusing and not clear. I suggest revising the methodology section to be more scientific and more powerful. you should draw a  figure reflecting the proposed method. also used an algorithm (or pseudo code) to explain the steps of the proposed Player Assessing

Concern # 5: Explain the system requirement. why did you select the inertial sensors? the number of selected sensors. the position of sensors and the reason for choosing these places? all this information must be explained in the context of the paper. 

Concern # 6: explain the performance metrics used in this study.

Concern # 7: section "2.2 Data Collection and Analysis" is so confused and not clear. I suggest to revised this section to be readable.

Concern # 8: The paper contains errors and typos. Remove the replicated sentences from the whole article and correct typos.

Concern # 9: The English level is low. I suggest proofreading by a specialist agent.

Evaluation and Results

Concern # 10: Any comparative analysis to testify that this study is more advanced than others? Discuss similar paper

Concern # 11: What are the defects of the method, and what are the methods that must be followed to reduce these defects?

Concern # 12: The discussion of the results is not satisfactory. Therefore, it is required to discuss the results scientifically to clarify claims and outcomes. 

Conclusion

Concern # 13: rewrite the conclusion and consider the following comments:

-   Highlight your analysis and reflect only the important points for the whole paper.

-   Mention the benefits.

-   Mention the implication at the last of this section.

Concern # 14: Try to include more than one direction for future work.

In my opinion, this paper is interesting and valuable, but some minor revisions may be necessary. Please carefully revise my comments.

Author Response

Point 1: Re-organize the abstract to include: (a) the overall purpose of the paper and the research problems you investigated should be short, (b) the basic design of the study, (c) major findings or trends as a result of the study, (d) a brief summary of your interpretation and conclusions.

Response 1: We have modified the abstract to make our purpose, design, findings, and interpretation clear.

Point 2: If the term is mentioned once in the introduction, there is no need to use abbreviations. In addition, when using the abbreviation, it must be written completely the first time.

Response 2: We apologize but did not find any abbreviations in our introduction. We have corrected the abbreviation format in the Figure 2 legend.

Point 3: Try to include additional keywords.

Response 3: We have included additional keywords.

Point 4: [Introduction] The paper needs to clarify the motivation, challenges, contribution, objective, significance, and others. All information should be presented in sequence idea. There is a need to answer the following questions: Q1: how can you evaluate the presented results according to other studies? Prove that the literature review lacks such a study by more modern references. Q2: What is the importance of the paper? Q3: What is the main challenge and issue in this study? Q4: What is the criticism and gap analysis for academic literature attempts to provide a solution? Q5: What are the recommended solutions for such challenges and their issues?

Response 4: We have edited the introduction and relevant (recent) citations to demonstrate the gap in understanding and the solution inertial sensors may provide in utilizing biomechanical factors for skill development.

Point 5: [Introduction] Explain the importance of Player Assessing by adopting it in many fields and applications. I suggested citing several previous works that used AI to solve different problems. Player Assessing as a literature review.

Response 5: See response 4. We have edited the introduction to explain the importance considering previous work.

Point 6: [Methodology] The proposed methodology is so confusing and not clear. I suggest revising the methodology section to be more scientific and more powerful. You should draw a figure reflecting the proposed method. Also used an algorithm (or pseudo code) to explain the steps of the proposed Player Assessing.

Response 6: We have revised the methodology section to be clearer.

Point 7: [Methodology] Explain the system requirement. Why did you select the inertial sensors? The number of selected sensors. The position of sensors and the reason for choosing these places? All this information must be explained in the context of the paper.

Response 7: Explanation and sensor details are included in Lines 112 – 132.

Point 8: [Methodology] Explain the performance metrics used in this study.

Response 8: Explanation of performance metrics are included in Lines 105 – 111.

Point 9: [Methodology] Section “2.2 Data Collection and Analysis” is so confused and not clear. I suggest revising this section to be readable.

Response 9: We have re-organized and edited section 2.2. to make it more readable.

Point 10: [Methodology] The paper contains errors and typos. Remove the replicated sentences from the whole article and correct typos.

Response 10: We have thoroughly reviewed the manuscript for typos or grammatical issues.

Point 11: [Methodology] The English is low. I suggest proofreading by a specialist agent.

Response 11: We have proofread the manuscript again. As English is our first language, we don’t feel a specialist agent is necessary.

Point 12: [Evaluation and Results] Any comparative analysis to testify that this study is more advanced than others? Discuss similar paper

Response 12: We have rewritten the discussion to highlight comparison of our study findings to existing literature.

Point 13: [Evaluation and Results] What are the defects of the method, and what are the methods that must be followed to reduce these defects?

Response 13: We have detailed the limitations of our study method and offered solutions to resolve these limitations (Lines 230 – 243).

Point 14: [Evaluation and Results] The discussion of the results is not satisfactory. Therefore, it is required to discuss the results scientifically to clarify claims and outcomes.

Response 14: We have rewritten the discussion to clarify our claims and outcomes.

Point 15: [Conclusion] Rewrite the conclusion and consider the following comments: (a) highlight your analysis and reflect only the important points for the whole paper, (b) mention the benefits, (c) mention the implications at the last of this section.

Response 15: We have re-written the discussion and conclusion based on the suggestions above.

Point 16: [Conclusion] Try to include more than one direction for future work.

Response 16: We have provided three directions for future research in connection with our study limitations: 1) increase and include biomechanical measures in relation to pitching skill, 2) use sensors to monitor athletes over time as they develop in skill level, and 3) analyze pitchers with varying skill levels.

I my opinion, this paper is interesting and valuable, but some minor revisions may be necessary. Please carefully revise my comments.

Reviewer 2 Report

In their Manuscript, the authors of the manuscript examined the pitching performance of ten athletes, using six inertial sensors and one pitch tracking device. The paper is intriguing, but the reporting in the current draft makes it difficult to read. In my opinion, there are too many tables in the supplemental materials (8 tables total; 1 table only in the paper), which renders the paper itself unsatisfactory. To make the Manuscript more readable, I advise thorough revision—especially with regard to the Results section that should be completely rearranged. Additionally, for the sake of clarity, I recommend that the statistical analysis section of the results follow the same order.

Other suggestions:

·       Please include at least one sentence describing the study's background and objectives in the abstract;

·       How can the pitchers' levels be described? Are they professionals?

·       Line 88: stride time and stride length are two completely different concept, please revise;

·       The pitch tracking device's user manual is not accessible via the provided link; please update it;

·       Do not include references to results in the statistical analysis section (see lines 166 - 168)

·       Please update the literature review as the references appear to be quite dated.

Author Response

Point 1: In their manuscript, the authors of the manuscript examined the pitching performance of ten athlete, using six inertial sensors and one pitch tracking device. The paper is intriguing, but the reporting in the current draft makes it difficult to read. In my opinion, there are too many tables in the supplemental materials (8 tables total; 1 table only in the paper), which renders the paper itself unsatisfactory. To make the manuscript more readable, I advise thorough revision – especially regarding the Results section that should be completely rearranged. Additionally, for the sake of clarity, I recommend that the statistical analysis section of the results follow the same order.

Response 1: We have rearranged the statistical analysis and results sections to match. We’ve removed any statistical analysis and results that were not relevant to our primary objective. As a result, were able to reduce the number of tables in the supplemental materials.

Please note that we also chose to remove the methods and results about curveball and spin rate. Command and ball velocity were our main outcomes, so we streamlined what was provided to readers.

Point 2: Please include at least one sentence describing the study’s background and objectives in the abstract: (a) how can the pitchers’ level be described? Are they professionals?

Response 2: We have edited the abstract to provide background and purpose. The level of the pitcher is described in the abstract (collegiate-level pitchers).

Point 3: Line 88: stride time and stride length are two completely different concept, please revise.

Response 3: We have revised the text.

Point 4: The pitch tracking device’s user manual is not accessible via the provided link; please update it.

Response 4: The pitch tracking user manual is no longer accessible online. We have added a copy to supplemental materials.

Point 5: Do not include references to results in the statistical analysis section (see lines 166-168).

Response 5: We have removed the reference.

Point 6: Please update the literature review as the references appear to be quite dated.

Response 6: We have updated the relevant literature in this manuscript.

Round 2

Reviewer 1 Report

Dear Authors

This manuscript is interesting and contains important information for readers. I can recommend this paper for publication with no modifications needed

Regards

Author Response

Thank you for your thoughtful and helpful comments. They have made the manuscript stronger.

Reviewer 2 Report

I would like to thank the Authors for having considered my suggestion. I only suggest including the tables, currently reported in the supplementary files, in the main text. I do not have any further suggestions.

Author Response

We have added the tables from the supplementary files to the main text.
